# Genetic/Protein Association of Atopic Dermatitis and Tooth Agenesis

**DOI:** 10.3390/ijms24065754

**Published:** 2023-03-17

**Authors:** Wanlu Ouyang, Charlene E. Goh, Wei Bo Ng, Fook Tim Chew, Eric Peng Huat Yap, Chin-ying Stephen Hsu

**Affiliations:** 1Department of Orthodontics, Shanghai Xuhui District Dental Disease Prevention and Control Institute, No. 500, Fenglin Road, Shanghai 200032, China; 2Faculty of Dentistry, National University of Singapore, Singapore 119085, Singapore; 3Faculty of Science, National University of Singapore, Singapore 117543, Singapore; 4Allergy and Molecular Immunology Laboratory, Lee Hiok Kwee Functional Genomics Laboratories, Department of Biological Sciences, Faculty of Science, National University of Singapore, Block S2, Level 5, 14 Science Drive 4, Lower Kent Ridge Road, Singapore 117543, Singapore; 5Lee Kong Chian School of Medicine, Nanyang Technological University, Singapore 636921, Singapore

**Keywords:** atopic dermatitis, tooth agenesis, skin barrier, gene–protein interaction

## Abstract

Atopic dermatitis and abnormalities in tooth development (including hypomineralization, hypodontia and microdontia) have been observed to co-occur in some patients. A common pathogenesis pathway that involves genes and protein interactions has been hypothesized. This review aims to first provide a description of the key gene mutations and signaling pathways associated with atopic dermatitis and tooth agenesis (i.e., the absence of teeth due to developmental failure) and identify the possible association between the two diseases. Second, utilizing a list of genes most commonly associated with the two diseases, we conducted a protein–protein network interaction analysis using the STRING database and identified a novel association between the Wnt/β-catenin signaling pathway (major pathway responsible for TA) and desmosomal proteins (component of skin barrier that affect the pathogenesis of AD). Further investigation into the mechanisms that may drive their co-occurrence and underlie the development of the two diseases is warranted.

## 1. Introduction

Atopic dermatitis (AD), also known as eczema and atopic eczema, is the most common chronic inflammatory skin disease [1] and is estimated to have the highest disease burden among all skin diseases [2]. 

Interestingly, some studies have shown epidemiological commonalities between AD and dental caries, and dental structural abnormalities such as hypomineralization and hypodontia (developmentally missing teeth). For example, in the GUSTO birth cohort study in Singapore, children diagnosed with AD in the first 18 months of life had a 3-times higher risk of developing tooth decay by age 3, despite controlling for several potential confounders [3]. A similar association was observed in adults in a nationwide cross-sectional study of 21,606 Korean adults, finding higher odds for having experienced caries in those with AD compared with those with no AD [4]. Another large population-based survey of Korean adolescents also showed significantly higher odds of AD among participants with oral symptoms, including sensitive teeth, toothache, etc. [5]. 

Due to the shared ectodermal tissue origin of the teeth and skin, an “ectodermal subclinical development defect” has been suggested, whereby genetic mutations associated with AD share a common pathogenic pathway with abnormalities in tooth development and can cause structural defects in the tooth, such as hypo-mineralization of the enamel. This structural defect in turn increases the tooth’s susceptibility to dental caries and may have resulted in the AD-caries associations observed. A longitudinal cohort of 6-year-old twins, which demonstrated moderate to strong associations between hypo-mineralization of the second molars (HSPM) and infantile eczema [6], provides further support for the link between AD and abnormalities in tooth development.

Tooth agenesis (TA) is an extreme case of abnormality in tooth development, where there is an absence of teeth due to developmental failure, and is one of the most prevalent dental and craniofacial malformations in humans [7]. TA can be categorized into the following three groups: hypodontia (less than 6 missing teeth), oligodontia (6 or more missing teeth) and anodontia (complete absence of dentition). 

Non-syndromic TA is TA that is not associated with any other systemic abnormalities or genetic syndromes. However, as the main cause of tooth agenesis is genetic, TA may involve other organs or tissues, as the involved networks of signaling molecules and transcription factors in TA and epithelial–mesenchymal interactions play essential and extensive roles during embryogenesis [8,9]. 

A few studies offer direct evidence for the association between AD and hypodontia. While only allergy (allergic rhinitis and pollinosis) was significantly associated, atopy (which includes atopic dermatitis) and asthma were also among the top conditions experienced by patients with hypodontia [10]. A 2017 Italian study found that 13/90 (14.4%) of children with atopic dermatitis had anatomical dental abnormalities, including agenesis and hypoplasias [11]. Our team also recently reported a significant association between severe–moderate AD and hypodontia and microdontia [12]. 

While the published review articles have focused on either AD or TA [13,14,15,16,17], there is an emerging need for a review that focuses on the overlapping areas and decodes the potential common signaling pathways and/or genes involved in the development of both AD and TA. 

Therefore, the aim of this review is to provide a description of the key gene mutations and signaling pathways associated with atopic dermatitis and tooth agenesis, respectively, identify the possible associations between the two diseases, and to propose exploratory hypotheses and mechanisms to narrow down the possible shared pathogenic pathways for future research to interrogate.

## 2. Genes Associated with Atopic Dermatitis (AD)

The two major pathophysiological pathways in AD are abnormalities of epidermal structure and function, and cutaneous inflammation due to inappropriate immune responses to antigens encountered in the skin [18]. Both pathways may influence each other and cause a systemic T helper type 2 (Th2) inflammatory pathway and a Th17/Th22 cell response, which may in turn affect epidermal structure and function. From the genetic point of view, the disease is inherited and multifactorial [19]. In this review, we will focus on the genes involved in epidermal barrier dysfunction, as both AD and TA are potentially caused by structural defects during ectodermal tissue development.

### 2.1. Mutations in Genes Related to Epidermal Barrier

The epidermal barrier is the first line of defense between the host organism and the environment. As illustrated in Figure 1, the skin barrier resides primarily in the *stratum corneum (SC)*, which consists of corneocytes surrounded by intercellular lipid lamellae and attached by *corneodesmosomes* [20]. The *tight junctions* attached to lateral walls of keratinocytes in the upper stratum granulosum (SG) have also been included in the basic skin barrier structure. Keratin filaments form macrofibrils by cross-linking with the *cornified envelope (CE)* of corneocytes. The SC lipid layer is covalently attached to the external surface of CE proteins, forming the cornified lipid envelope (CLE) [21].

Stratum corneum (SC): The most significant gene variants associated with AD are the loss-of-function mutations found in the filaggrin (FLG) gene. An estimated 27.5% of Caucasian Americans, 48% of Europeans, 31.4% of Chinese and 20% of Japanese populations with AD present mutations in the FLG gene [22]. Filaggrin is a protein found in the corneocytes responsible for aggregating keratin in the formation of the stratum corneum [23]. It is produced from a precursor, pro-filaggrin. Studies have confirmed that FLG null mutations increase the risk of AD, impairing skin barrier function [24,25]. Homozygous mutations in the FLG gene are associated with an increased risk of severe AD, with earlier onset, longer duration, and increased skin infections [26,27,28]. In addition, filaggrin has a broad range of immunomodulatory effects [28,29,30,31].

Filaggrin-2 (FLG-2) is a filaggrin-like protein, and is part of the corneocyte envelope [32]. The expression of the FLG2 protein is reported to be decreased in patients with AD [33]. A link between polymorphisms in the FLG2 gene and more persistent AD in African American populations has been found [34]. However, a recent study in Brazil found no correlation between AD patients and polymorphisms in the FLG2 gene [35].

OVOL1 (ovo-like transcriptional repressor) is an upstream transcription factor that regulates FLG expression. FLG, OVOL1 and IL13 are reported to be the three genes most significantly associated with AD among the 31 susceptible gene loci reported in a meta-analysis of genome-wide association studies [36].

Other gene mutations that result in epidermal barrier dysfunction include genes that encode SC proteases such as KLK7, and genes that encode protease inhibitors SPINK5 and LEKTI. Kallikreins are a family of 15 trypsin- or chymotrypsin-like secreted serine pro-teases (KLK1-KLK15). The expression of KLK5-8, KLK10, KLK13 and KLK14 is significantly increased in AD patients [37], and the elevation of KLK7 is predominant in the SC [37]. A 4-bp insertion in the 3’-untranslated region of the KLK7 gene was found to have a significant association with AD [38].

The multidomain serine protease inhibitor Kazal-type 5 (SPINK5), otherwise known as the lympho-epithelial Kazal-type-related inhibitor (LEKTI), plays a role in keratinocyte differentiation during skin and hair morphogenesis, and the protective barrier function of skin by inhibiting the activity of KLKs in the epidermis [39,40]. Loss of KLK regulation through SPINK5 mutation could cause excessive KLK activity, resulting in permeable barrier defects [40,41]. AD has been associated with SPINK5 mutations in certain populations, specifically eastern Asians [40,42,43,44,45,46].

Cornified envelope (CE): The corneocyte envelope (CE) serves as a scaffold for lipids to attach and provides a supportive force to the corneocytes. The envelope is formed from structural proteins, including involucrin, loricrin, and the small proline-rich (SPRR) proteins [47]. An extra 24-bp defect in the central domain and additional in-frame deletions and insertions of the SPRR3 gene have been associated with AD [48]. Levels of FLG, FLG2, and SPRR3 mRNAs and proteins were also found to be reduced in AD skin [49]. 

Corneodesmosomes and tight junctions: Mutations within genes that express corneodesmosomal proteins (desmogleins and desmocollins) and tight junction proteins (claudins and ocludins) also contribute to the progression of AD [43]. Desmoglein-1 (DSG1), Claudin-1 (CLDN1) and Claudin-23 (CLDN23) have been reported to cause downregulation in AD [50]. CLDN1 haplo-type-tagging single nucleotide polymorphisms reveal linkage to AD in two North American populations [51]. The risk of eczema herpeticum in AD subjects is also associated with variants in the CLDN1 gene [51]. DSG3-/- mice appear to have traumatized skin that displays a distinct separation of desmosomes under electron microscopy [52]. Mice lacking desmocollin 1 (DSC1) have a fragile and flaky epidermis with acanthosis in the stratum granulosum [53].

### 2.2. Gene Polymorphisms in Inflammation and Immunity

Genetic variants associated with these immunological events contribute to the aberrant inflammatory and immune response in AD. Mutations in pattern-recognition receptors (PRRs) have been observed to be related to AD; these include polymorphisms in toll-like receptors (TLRs)—TLR2, TLR4, TLR6, TLR9—and gene polymorphisms in nucleotide-binding oligomerization domain-like receptors (NLRs)—CARD4, CARD12, CARD15, NALP1, NALP12, and NOD1; several SNPs of the human β-defensin 1 (DEFB1) gene have also been found in AD patients [19]. Mutations in IL-1 family cytokines and receptors genes that induce systemic Th2-type inflammatory responses, e.g., the susceptibility loci 2q12, which contain the receptors of the IL-1 family cytokines (IL1RL1, IL18R1, and IL18RAP) and the IL-18 gene play key roles in innate immunity and contribute to the pathogenesis of AD. Mutations in genes implicated in the vitamin D metabolism and synthesis of its receptors (CYP27A1, CYP2R1 and VDR) have been reported to be associated with AD. Mutations in interleukin genes produced by keratinocytes, including IL-25, TSLP, IL-33 and IL-7RA, were found in the epidermis in lesions of AD exposed to stress, e.g., UV or mechanical trauma. The adaptive immune response in AD is associated with the increased expression of the Th2 cytokines (IL-4, IL-5, IL-13, and IL-31) and the Th22 cytokine IL-22 during the acute phase of AD [54,55]. Several distinct polymorphisms of IL-4, IL-5, IL-13, IL-4 receptor alpha (IL-4RA), IL-5 receptor alpha (IL-5RA), and IL-13 receptor alpha (IL-13RA) have been found to influence the susceptibility to AD in different populations. Genetic variants in IL-12 and IL-12R [54,55], IFNG and IFNGR1, as well as interferon regulatory factor (IRF)-2, were significantly associated with AD and eczema herpeticum (EH) [56,57]. Other cytokine and receptor variants were also identified in AD, including IL-2, IL-6, IL-9, and IL-10 [19]. Correlations between AD and genetic polymorphisms of FcεRIα—the alpha-chain of high-affinity IgE receptors—have also been observed [19].

## 3. Genes Associated with Tooth Agenesis (TA)

Tooth development is a series of genetically regulated processes with successive and reciprocal interactions of the epithelium and mesenchyme (Figure 2). Four major signaling pathways (Fgf, Wnt, Bmp and Shh) and numerous transcription factors are key to tooth development. Disturbances at any stage or alterations in any pathway may lead to tooth agenesis [58].

### 3.1. Paired Box Gene 9 (PAX9)

PAX9 is the most prevalent gene for non-syndromic TA [60]. It encodes a member of the paired box family of transcription factors and is expressed in the mesenchyme to induce odontogenic signals and initiate dental development [61]. PAX9 is important for organogenesis, as it induces the activation of Wnt and TGF-β/BMP signaling pathways [62]. Mutations in PAX9 often lead to the absence of second molars [61,63,64], and are associated with a high risk of maxillary lateral incisor agenesis [65]. Second premolars were also sometimes affected [7,66]. 

### 3.2. Muscle Segment Homeobox 1 (MSX1)

MSX1, a member of the homeobox genes, encodes for a protein that acts as a transcriptional repressor during embryogenesis and is critical for the development of teeth [67]. The Wnt/β-catenin signaling may increase MSX1 expression which subsequently activates the TGF-β/BMP cascade for odontogenesis [68]. Mutations in MSX1 have been associated with severe forms of hypodontia, oligodontia with cleft lip, and non-syndromic TA, usually missing mandibular central incisors, upper lateral incisors, second premolars, and third molars [61,69,70,71]. Regulation of BMP4 expression can also be affected through the synergistic interaction between MSX1 and PAX9 [72].

### 3.3. Axis Inhibitor 2 (AXIN2)

AXIN2 encodes an intracellular inhibitor of Wnt/β-catenin signaling and has been associated with lower incisor agenesis [73,74]. AXIN2 missense mutants were found to enhance β-catenin degradation and reduced Wnt activation, whereas the truncated mutants seemed to heighten the activation of Wnt/β-catenin [75]. A nonsense mutation in the AXIN2 gene was reported as etiologic for familial TA and predisposes patients to colorectal cancer [76]. AXIN2 is highly expressed in the enamel knot and underlying mesenchyme during tooth formation in mice [76]. 

### 3.4. Ectodysplasin A (EDA) and Relevant Genes

Ectodysplasin A, a protein of the tumor necrosis factor family, plays an important role during the development of ectodermal organs and teeth by activating the IKBKG-NF-κB signaling pathway. Mutations in EDA and EDA receptor genes have been reported to affect sporadic hypodontia in families [77]. Most of the mutations in EDA are identified to cause X-linked hypohidrotic ectodermal dysplasia (HED) [78]. In addition, some of these EDA mutations have also been associated with missing maxillary lateral incisor cases [64].

Most mutations in EDAR and EDARADD are associated with ectodermal dysplasia, with a few associated with non-syndromic TA. Several mutations of IKBKG will lead to incontinentia pigmenti and ectodermal dysplasia (HGMD); therefore, NF-κB activity is likely to be affected by these mutations in the aforementioned genes [60].

In addition, single nucleotide polymorphisms in the EDAR gene have also been associated with other dental malformations. For example, the presence/absence of the V370A allele of the EDAR gene has been correlated with modern human shovel-shaped incisors [79]. The 1540C allele of EDAR was also found to be strongly associated with the presence of incisor shoveling and hair thickness [80].

### 3.5. Other Genes Related to Wnt Signaling Pathway

Recently, the genetic link between the Wnt pathway and TA was highlighted once again through whole exome and Sanger sequencing, with the observation of many mutations in genes that encode for Wnt ligands and its receptors [60]. The reported genetic mutations include genes that encode for Wnt ligands such as WNT10A and WNT10B, and associated receptors such as LRP6. For individuals with non-syndromic TA, WNT10A is one of the most commonly mutated genes [60]. It is expressed in the dental epithelium at the dental lamina and bud stage and in the enamel knot during the cap stage [81]. Mutations in WNT10A account for more than half of the isolated hypodontia and oligodontia cases [81,82] and have been identified with odonto-onycho-dermal dysplasia [83,84,85].

Wnt10B, a structurally related protein, is also expressed in the dental epithelium during the early bud and cap stages of tooth development. Similar to Wnt10A, genetic mutations in Wnt10B have been found in dental anomalies, such as TA and oligodontia [86,87]. Impaired odontoblastic differentiation and vasculogenesis of dental stem cells can result from these Wnt10B mutants, as they are unable to activate Wnt signaling pathways [87].

LRP6 is important for cell differentiation and proliferation, as it encodes a protein of the Wnt-Fzd-LRP5-LRP6 complex, which triggers the Wnt/β-catenin signaling cascade. LRP6 mutants are unable to activate β-catenin, therefore preventing the activation of Wnt signaling [87]. These LRP6 mutations have been reported in those with non-syndromic TA [88].

Other genes implicated in isolated TA or oligodontia accompanied with ectodermal dysplasia include mutations in DKK1 and associated KREMEN1; while mutations in ANTXR1 were also implicated with syndromic TA [89,90,91]. DKK1, Dickkopf Wnt signaling pathway inhibitor 1, is involved in the regulation of embryonic and vascular development when it binds to the transmembrane receptor, KREMEN1, and co-receptor, LRP6, to inhibit Wnt/β-catenin signaling. Through its interaction with LRP6, ANTXR1 plays an important role in modulating Wnt signaling and stabilizing β-catenin [60].

### 3.6. Other Genes Related to TGF-β/BMP Signaling Pathway

The implicated TGF-β/BMP-associated genes include GREM2—for mutations associated with TA, taurodontism, short tooth roots, and microdontia—and LTBP3—for mutations associated with inherited dental anomalies and isolated oligodontia [92,93,94]. GREM2 is involved in the regulation of embryonic morphogenesis, specifically TGF-β signaling in tooth development, as it encodes for a BMP antagonist protein [91]. LTBP3 encodes for a protein that regulates the assembly, secretion, and targeting activity of the TGF-β molecules, through the formation of a complex [93,94].

### 3.7. SMOC2 Gene

Secreted protein acidic and rich in cysteine (SPARC)-related modular calcium binding 2 (SMOC2) is an early dental developmental gene in human beings, supported by its high expression in areas such as the oral ectoderm and dental epithelium [95]. This gene encodes a member of the SPARC family protein, which promotes matrix assembly and stimulates endothelial cell proliferation and migration, as well as angiogenic activity [96]. Dental anomalies such as dental dysplasia, severe oligodontia and extreme microdontia have been reported with SMOC2 gene mutations [97].

## 4. Protein–Protein Interaction Network Functional Enrichment Analysis in AD and TA

From the above review of the gene mutations associated with epidermal barrier defects in AD and TA, no direct overlap of the genes involved was observed. Nevertheless, it is possible that there are shared biological pathways or processes, or indirect interactions through intermediary molecules between the two diseases. Recent research has shown that protein–protein interactions (PPI) are crucial for most biological activities, and examining the protein interaction networks can help to identify key proteins that may be involved in both diseases or that act as “hubs” linking multiple pathways. For example, PPIs between Parkinson’s disease and periodontitis have been found to indicate new candidate molecular mechanisms [98]. Therefore, we aim to conduct a protein interaction analysis between AD and TA to explore new potential targets for research purposes.

### 4.1. Methods

The protein–protein interaction networks for TA and AD were investigated using version 11.5 of the STRING (Search Tool for the Retrieval of Interacting Genes/Proteins) database (https://string-db.org/, accessed on 15 November 2022), together with association and analysis methods. The STRING database contains known and predicted protein–protein interactions, stemming from computational prediction, knowledge transfer between organisms, and interactions aggregated from other (primary) databases. In general, PPIs in STRING are derived from the following five main domains: Genomic Context Predictions, High-throughput Lab Experiments, (Conserved) Co-Expression, Automated Textmining and Previous Knowledge in Databases (https://cn.string-db.org/cgi/about?footer_active_subpage=content, accessed on 15 November 2022). Apart from in-house predictions and homology transfers, STRING also relies on many resources maintained elsewhere (https://cn.string-db.org/cgi/credits?footer_active_subpage=datasources, accessed on 15 November 2022). The methodological details of the STRING database and network analysis have been reported in a recent paper [99]. 

The lists of proteins associated with AD and TA as summarized above and in Figure 1 and Figure 2 were input and a “high confidence” cutoff of 0.7 was set, as is the case in previously published studies [98,100,101]. A PPI network was then generated [102], as shown in Figure 3.

### 4.2. Results

In the resulting protein–protein interaction network map, proteins are presented as nodes, which are connected by color-coded lines representing potential protein–protein associations. We identified potential protein–protein interactions (PPIs) between the proteins known to be associated with TA and AD. Catenin beta-1 (CTNNB1) presents interactions with TA-associated genes, including AXIN2, WNT10A, WNT10B and LRP6, as a key downstream component of the canonical Wnt signaling pathway. Importantly, there is evidence that indicates functional links between CTNNB1 and desmosomal proteins (DSC1 and DSG3), which play an important role in the maintenance of skin barrier function, and thus affect the pathogenesis of AD (Figure 3). There are “experimentally determined” interactions (denoted by a pink line) between CTNNB1 and DSC1, and other interactions derived from “text-mining”, meaning “co-mentioned in PubMed abstracts” (denoted by a light green line) and co-expression of genes (denoted by a black line). Furthermore, between CTNNB1 and DSG3, there are known “experimentally determined” interactions and “text-mining” interactions.

## 5. Discussion and Limitations

While the Wnt/b-catenin pathway to junctional/desmosomal proteins interaction may have been previously determined experimentally (as illustrated by the purple/pink connection line), this interaction has not been proposed to be involved in the mechanistic pathways through which AD and TA co-occur. The identification of the interaction of these proteins that are associated with the known gene mutations for these two diseases is the novel result of this database search.

The β-catenin protein encoded by the CTNNB1 gene is part of a complex of proteins that constitute adherens junctions (AJs). AJs are necessary for the creation and maintenance of epithelial cell layers by regulating cell growth and adhesion between cells. Meanwhile, β-catenin is an integral part of the canonical Wnt signaling pathway.

Tooth development is a dynamic process that goes through the bud, cap and bell stages, root development and tooth eruption [103]. The Wnt/β-catenin signaling pathway is involved in embryonic development in many aspects and is active at all stages in various regions of tooth-forming, playing a key role in odontogenesis [104]. The machinery of the Wnt pathway includes extracellular secreted glycoproteins (19 Wnt ligands at the human level), seven transmembrane-spanning receptors (Frizzled and LRP5/6), cytoplasmic proteins (DVL, APC, AXIN, GSK3β and β-catenin, etc.), nuclear transcription factors (TCF/LEF), and several related molecules (MSX1, DKK1, KREMEN1, and ANTXR1) [60]. The Wnt and Wnt-associated pathways are demonstrated to play a major role in the molecular pathogenesis of the non-syndromic TA [60].

Research on the relationship between desmosomal components and β-catenin signaling has been carried out. It was shown that DSC3 regulated β-catenin in suprabasal keratinocytes by inducing β-catenin stabilization and transgene-mediated DSC3a and DSC3b expression in differentiating keratinocytes enhances β-catenin signaling [105]. The abnormal expression of DSC3, DSG3, and β-catenin was found in oral carcinomas and that the reduced or absent expression of β-catenin had a positive correlation with reduced or absent expression of DSC3 in 24 patients with lymph node metastasis [106]. Silencing DSG3 was reported to inhibit the activation of the Wnt/β-catenin signaling pathway in mice with chronic rhinosinusitis [107]. Additionally, Sawa et al. found that human dental pulp fibroblasts did not express desmoplakin (DPK, cytoplasmic membrane-associated protein in the epithelium) until they were cultured in the differentiation medium, whereas odontoblasts expressed vimentin-binding DPK-1 [108].

The canonical Wnt/β-catenin pathway is a fundamental mechanism that accounts for various biological activities, including cell proliferation, differentiation and development. Distinct from the non-canonical Wnt pathways that are independent of β-catenin, the canonical Wnt/β-catenin pathway has β-catenin as its typical characteristic [109]. In the inactive state, the β-catenin protein is degraded by a destruction complex composed of AXIN, glycogen synthase kinase 3β (GSK3β) and adenomatous polyposis coli (APC) [109]. Upon Wnt ligand receptor/coreceptor (LRP5/6) binding, this complex becomes inactivated, leading to cytoplasmic accumulation and subsequent nuclear translocation of β-catenin [110]. In the nucleus, β-catenin stimulates the transcription of target genes in cooperation with T-cell-factor/lymphoid enhancer-binding factors to regulate the expression of downstream target genes [111]. The effects of the Wnt/β-catenin inhibitor ICG-001 in an AD-like murine model generated by repeated topical application of the hapten oxazolone (Ox) were examined, and ICG-001 was found to attenuate epidermal permeability barrier function in Ox-AD mice [111]. A longitudinal birth cohort study that involved 1699 children in Korea found that children with the AD phenotype in early life were closely related to the development of asthma only in the cases of accompanying food allergy (FA) [112]. Ingenuity pathway analysis (IPA) of the colonocyte transcriptome revealed that the differentiation of FA with AD was best described by the genes in ‘Wnt/β-catenin Signaling’, specifically AXIN1, CCND1, FZD2, and WNT6, indicating that this mechanism might be regulated by Wnt signaling [112]. This pathway also has a strong link with TA.

In summary, PAX9 and MSX1 are the most prevalent genes for non-syndromic TA. Mutated genes that encode the components in the canonical Wnt/β-catenin pathway and Wnt-associated genes account for the highest genetic risk for isolated TA compared to mutated genes involved in several other pathways. TGF-β/BMP and EDA/EDAR/NF-κB signaling pathways also contribute to TA. Skin epithelial function and immune responses are the two major biologic pathways responsible for AD and they interactionally affect each other. Genetic mutations that cause structural defects in the epidermal barrier include genes that encode epidermal barrier structural proteins, stratum corneum proteases and protease inhibitors.

We identified protein interactions between desmosomal proteins and β-catenin. Desmosomes and tight junctions are essential components of the skin barrier, affecting the pathogenesis of AD. β-catenin is a key component of the Wnt/β-catenin signaling pathway, which is the major pathway responsible for TA. While experimental studies have shown that the DSG and DSC levels have a positive correlation with β-catenin expression and affect the Wnt signaling pathway, this protein interaction has not been previously proposed as one of the mechanistic pathways through which AD and TA co-occur. The specific mechanism by which these two diseases interact through this pathway remains unclear and more evidence is needed, for example, the GWAS risk of disease and EQTL analysis. Nevertheless, our findings help to narrow down the possible shared pathogenic pathways for future research to interrogate in this novel field, and help support the hypothesis that shared genetic mutations in the epidermal structure could increase the risk of tooth agenesis, thus linking structural defects in the skin barrier and tooth formation. Further studies may explore the mechanisms of gene variation in desmosomes and adherens in epithelial membranes, and their interactions with the Wnt signaling pathway linked to TA, in particular in the context of patients with TA and AD.

## Figures and Tables

**Figure 1 ijms-24-05754-f001:**
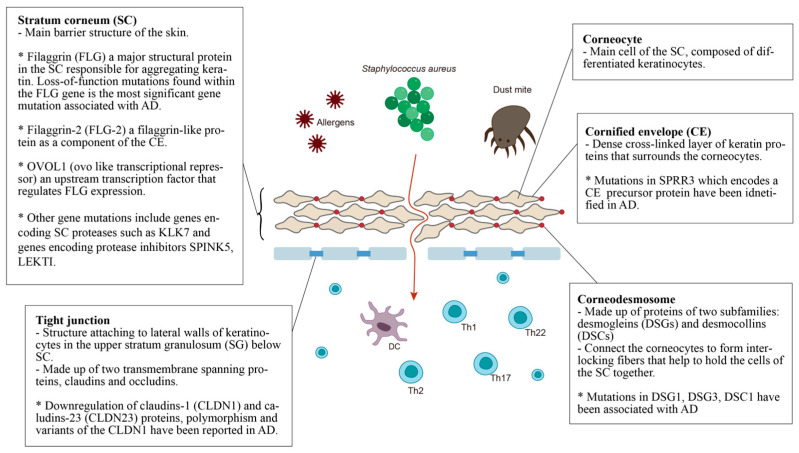
Defective epidermal barrier in atopic dermatitis (AD). The genes that affect SC may have an effect on both corneocytes and CE. For example, the filaggrin protein can be found in corneocytes but it aggregates keratin, affecting the whole SC. Therefore, genes with specific localization of action are placed in the small boxes, while the others are placed in the larger box of SC.

**Figure 2 ijms-24-05754-f002:**
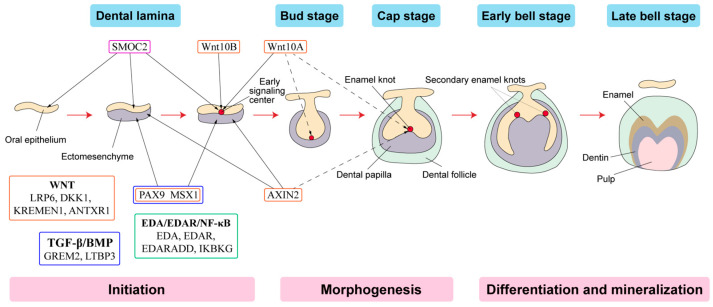
Schematic representation of tooth development stages. (Modified from the work of Nanci A. in 2013. Development of the tooth and its supporting tissues [59]) Main groups of signaling pathways and genes associated with tooth agenesis (TA). Gene mutations in Wnt pathway (orange box), TGF--β/BMP pathway (blue box), EDA/EDAR/κB pathway (green box) and SPARC family (purple box) affecting the initiation stage of tooth development results in tooth agenesis.

**Figure 3 ijms-24-05754-f003:**
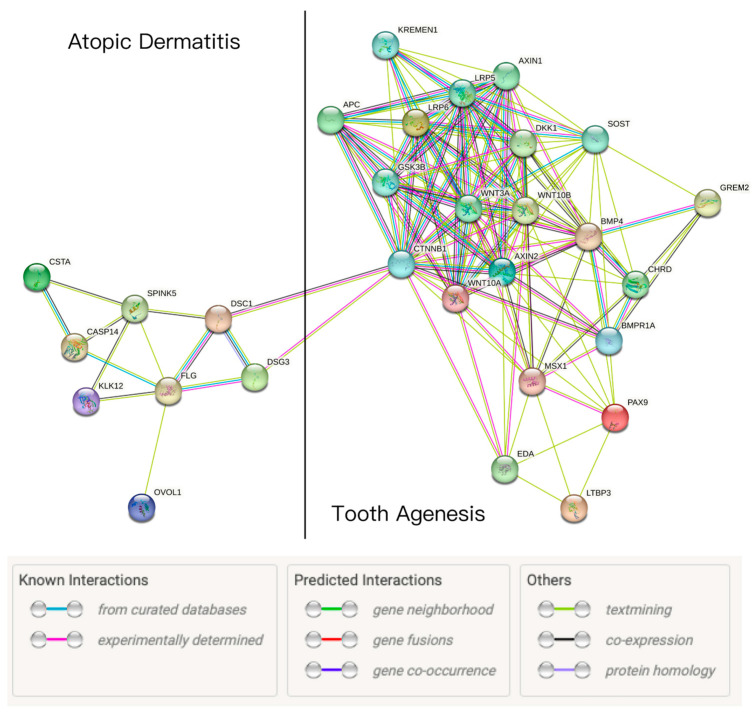
Search Tool for the Retrieval of Interacting Genes/Proteins (STRING) analysis reveals protein interaction networks between tooth agenesis and atopic dermatitis proteins. The color-coded nodes presented proteins, while color-coded lines represent protein–protein associations. Light blue lines represent associations in curated databases; pink lines represent experimentally determined associations derived from experimental/biochemical data; light green lines represent text-mining associations meaning “co-mentioned in PubMed abstracts”; black lines represent co-expression of genes in homo sapiens or other organisms; light purple lines represent putative homologs. There were no predicted interactions (green, red and blue lines) in the results.

## Data Availability

Not applicable.

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
