# Peer review of "Genetic/Protein Association of Atopic Dermatitis and Tooth Agenesis"

_ijms, 2023, doi:10.3390/ijms24065754_

Round 1

Reviewer 1 Report

Oyang et al. give an overview of candidate genes/proteins that might commonly cause both atopic dermatitis (AD) and tooth agenesis (TA). This is an interesting clinical observation, that deserves a closer look.

They review the literature for the single diseases (with littel overlap) and then perform a database run on protein interactions, that results in a map. It is not completely clear, what is feeded into the calculation and what is the meaning of topological neighborhood in the resulting map.

Content, structure, references and language are fine, but:

The „resulting“ vicinity of the Wnt/b-catenin pathway to junctional/desomosomal proteins is well established experimentally (as depicted by the purple colour in the connection lines). What is the new result of this database search? The main concern is on the take-home message, which so far is lacking or trivial.

Minor:

l. 33: „commonalities“ is not existing
l. 45 „bolstered“ would be better expressed by „supported“
l. 136/7 „so-ciated“ „ma-tized“
l. 212 „some…has“ (plural)
l. 346 „gene_“ should read plural
l. 351 „may affect each other“ and l. 358 „can have positive correlation“ is too weak. Please be more specific: Are there hints, then state: „there (most likely) is a correlation“ or omit (see main concern)

Author Response

Dear Editors and reviewers:

Thank you very much for the comments and suggestions, which we found relevant and beneficial for improving the quality of this paper.

Please find below our response, point by point, to the issues raised. The manuscript has been revised accordingly, with all references checked to be relevant to the contents of the manuscript and revisions marked up using the “Track Changes” function (as attached).

Reviewer 1

Point-1. They review the literature for the single diseases (with little overlap) and then perform a database run on protein interactions, that results in a map. It is not completely clear, what is feeded into the calculation and what is the meaning of topological neighborhood in the resulting map.

Authors’ Response:

Protein–protein interaction (PPI) networks have been used to identify genes that are significantly involved in specific PPI. Our study utilized STRING, a database containing known and predicted protein-protein interactions, stemming from computational prediction, knowledge transfer between organisms, and interactions aggregated from other (primary) databases. In general, PPI in STRING are derived from five main domains: Genomic Context Predictions, High-throughput Lab Experiments, (Conserved) Co-Expression, Automated Textmining and Previous Knowledge in Databases (https://cn.string-db.org/cgi/about?footer_active_subpage=content). Apart from in-house predictions and homology transfers, STRING also relies on many resources maintained elsewhere (https://cn.string-db.org/cgi/credits?footer_active_subpage=datasources). The methodological details were elaborated in a recent paper [1].

The names of genes expressing protein variants and families of proteins related to Tooth Agenesis (TA) and Atopic Dermatitis (AD) are listed below:

For TA: PAX9, MSX1, AXIN2, EDA, EDAR, EDARADD, WNT10A, WNT10B, LRP6, KREMEN1, DKK1, ANTXR1, GREM2, LTBP3, SMOC2

For AD: Filaggrin, kallikrein, OVOL1, SPINK5, LEKTI, desmoglein, desmocollin, claudin, ocludin, SPRR3

After inputting the lists of proteins associated with AD and TA, we set a “high confidence” cutoff of 0.7 as done in previously published studies [2-4] and obtained the results shown in Figure 3.

In the resulting protein-protein-interaction network map created, the color-coded nodes represent proteins, and color-coded lines represent potential protein-protein associations. The topographical neighborhood or layout of the nodes is determined by a graph layout algorithm that aims to optimize the visual arrangement of the nodes. The line colors denote the type of association, such as “experimentally determined interactions” (pink line), “text-mining interactions” (light green line), and “co-expression (black line). Shown in the figure 3, there are known “experimentally determined” interactions (denoted by a pink line) between CTNNB1 and DSC1, and other interactions derived from “text-mining” co-mentioned in PubMed abstracts (denoted by a light green line) and “genes co-expression” (denoted by a black line). Furthermore, between CTNNB1 and DSG3, there are known “experimentally determined” interactions and “text-mining” interactions as well.

The length of the lines (such as the longer lines between CTNNB1 and DSC1/DSG3) are simply used to visually connect the two nodes and do not have any correlation to the strength of the evidence.

We have amended the Methods section to include a summary of the description above, clarifying how we performed PPI.

References

  1. Szklarczyk, D.; Kirsch, R.; Koutrouli, M.; Nastou, K.; Mehryary, F.; Hachilif, R.; Gable, A. L.; Fang, T.; Doncheva, N. T.; Pyysalo, S.; et al. The STRING database in 2023: protein-protein association networks and functional enrichment analyses for any sequenced genome of interest. Nucleic Acids Res 202351(D1), D638-D646. DOI: 10.1093/nar/gkac1000.
  2. Botelho, J.; Mascarenhas, P.; Mendes, J. J.; Machado, V. Network Protein Interaction in Parkinson's Disease and Periodontitis Interplay: A Preliminary Bioinformatic Analysis. Genes (Basel) 202011(11). DOI: 10.3390/genes11111385.
  3. Leira, Y.; Mascarenhas, P.; Blanco, J.; Sobrino, T.; Mendes, J. J.; Machado, V.; Botelho, J. Network Protein Interaction in the Link between Stroke and Periodontitis Interplay: A Pilot Bioinformatic Analysis. Genes (Basel) 202112(5). DOI: 10.3390/genes12050787.
  4. Liu, T.; Wang, S.; Wornow, M.; Altman, R. B. Construction of disease-specific cytokine profiles by associating disease genes with immune responses. PLoS Comput Biol 202218(4), e1009497. DOI: 10.1371/journal.pcbi.1009497.

Point-2a. The resulting vicinity of the Wnt/b-catenin pathway to junctional/desomosomal proteins is well established experimentally (as depicted by the purple colour in the connection lines). What is the new result of this database search? The main concern is on the take-home message, which so far is lacking or trivial.

Response:

While the Wnt/b-catenin pathway to junctional/desomosomal proteins interaction may have been previously determined experimentally (as illustrated by the purple/pink connection line), this interaction has not been proposed as involved in the mechanistic pathways through which AD and TA co-occurs. The identification of the interaction of these proteins which are associated with the known gene mutations for these two diseases is the novel result of this database search, and further investigation of these interactions in the context of patients with TA and AD may be warranted. The take-home message is that despite a lack of direct overlap in the gene mutations and signaling pathways of AD and TA, there may be some protein-protein interactions, specifically between CTNNB1 and desmosomal proteins (DSC1 and DSG3) that link the two diseases. Our findings help narrow down the possible shared pathogenic pathways for future research to interrogate in this novel field.

In addition, the results indicated plausible interactions derived from text-mining (denoted by a light green line) and co-expression, in addition to the “experimentally determined” interactions between CTNNB1 and DSC1/DSG3. This aforementioned PPI has not been reported before. In addition to the associations that have been experimentally confirmed, there are some indirect connections that have not been experimentally confirmed yet. Furthermore, we wanted to dig more into the specific effects and potential associations these genes/proteins may jointly play in AD and tooth development to inform future researchers in this novel field.

Point-2b: Minor:

l. 33: „commonalities“ is not existingl. 45 „bolstered“ would be better expressed by „supported“l. 136/7 „so-ciated“ „ma-tized“l. 212 „some…has“ (plural)l. 346 „gene_“ should read plural

  1. 351 „may affect each other“ and l. 358 „can have positive correlation“ is too weak. Please be more specific: Are there hints, then state: „there (most likely) is a correlation“ or omit (see main concern)

All the minor issues raised by the reviewer-1 have been corrected in the revised manuscript. We have also made the suggested moderate language changes to our manuscript. Specifically, we rephrased some sections, and corrected the grammar and spelling errors throughout the paper. We believe that these changes have significantly improved the clarity and readability of our manuscript.

Reviewer 2

Point-3. The title of review paper “Genetic/Protein Association of Atopic Dermatitis and Tooth Agenesis” suggests that there are some associations between these two diseases, however there were not found. There are no citations of clinical or epidemiology papers which can suggest these association. Authors citate a few papers which indicate the coexisting AD and caries, oral symptoms. Authors cite the publication “Allergy as a possible predisposing factor for hypodontia” T. Yamaguchi et al, however in this paper allergy includes allergic rhinitis and pollinosis, no AD or eczema. I would like to know why authors perform this genetic analysis if there are no clinical or epidemiology reports about these associations. I think it is too far-reaching hypothesis without any proofs.

Response:

We appreciate the reviewer’s comment and agree that the interplay between AD and TA is still not well understood. However, a few studies offer some evidence for a potential association between the two conditions. Our team recently reported the novel link between hypodontia, (and microdontia) with moderate–severe AD [1]. In addition, a 2017 Italian study found 13/90 (14.4%) of children with atopic dermatitis exhibiting anatomical dental abnormalities including agenesis and hypoplasia [2].

In the Yamaguchi paper, while the only condition significantly associated with hypodontia was allergy (allergic rhinitis and pollinosis), atopy (which include atopic dermatitis) and asthma were also the top conditions experienced by patients with hypodontia. Following the “ectodermal subclinical developmental defect” (“ESDD”) hypothesis, our observation of a significant "AD-caries" association in the GUSTO birth cohort study, despite controlling for several potential confounders [3], led us to hypothesize that patients with hypodontia and AD may share genetic and/or developmental predisposing factors. Therefore, we explored this hypothesis in the current manuscript. The result of this study, featured in Figure 3, offer preliminary support for this hypothesis, and suggest potential direction(s) for future clinical, epidemiological, and laboratory studies to confirm the mechanistic explanation of AD-hypodontia comorbidity.

We have added the above examples and citations to our manuscript to clarify why we believe further examination of the “ESDD” hypothesis through a review of the genes/proteins involved in TA and AD, and an exploratory analysis of protein-to-protein interactions, was warranted.

References

  1. Tan, S.; M. Leong, S.; Hsu, C.-y.; Chandran, N. Association of moderate–severe atopic dermatitis with dental anomalies. Indian Journal of Dermatology 202267(5), 539-542, Short Communication. DOI: 10.4103/ijd.ijd_375_22.
  2. Perugia, C.; Saraceno, R.; Ventura, A.; Lorè, B.; Chiaramonte, C.; Docimo, R.; Chimenti, S. Atopic dermatitis and dental manifestations. G Ital Dermatol Venereol 2017152(2), 122-125. DOI: 10.23736/s0392-0488.16.05224-x
  3. Kalhan TA, Loo EXL, Kalhan AC, Kramer MS, Karunakaran B, Un Lam C, et al. Atopic dermatitis and early childhood caries: results of the GUSTO study. J Allergy Clin Immunol. 2017;139:2000–2003

Please feel free to clarify if there is anything not so clear.

Thanks again for all your efforts in making this paper of scientific value to the readers and scientific community.

Best regards,

Associate Professor Chin-Ying Stephen Hsu :: DDS, MS, PhD

Department of Dentistry, Faculty of Dentistry, 9 Lower Kent Ridge Road, National University Centre for Oral Health, Singapore, Singapore 119085 :: (65)-6772 6832 (DID) :: (65)-67732602 (Fax) :: [email protected] (E) :: www.nus.edu.sg (W), a member of NUHS, National University of Singapore ::  Company Registration No: 200604346E

Reviewer 2 Report

The title of review paper “Genetic/Protein Association of Atopic Dermatitis and Tooth Agenesis” suggests that there are some associations between these two diseases, however there were not found. There are no citations of clinical or epidemiology papers which can suggest these association. Authors citate a few papers which indicate the coexisting AD and caries, oral symptoms. Authors cite the publication “Allergy as a possible predisposing factor for hypodontia” T. Yamaguchi et al, however in this paper allergy includes allergic rhinitis and pollinosis, no AD or eczema. I would like to know why authors perform this genetic analysis if there are no clinical or epidemiology reports about these associations. I think it is too far-reaching hypothesis without any proofs. 

Author Response

(The authors gave the same response as above.)

Round 2

Reviewer 2 Report

Thank you very much for your detailed explanation. The work can be preliminary support for the hypothesis, that patients with hypodontia and AD may share genetic and/or developmental predisposing factors, however I am not fully convinced that the work is a significant contribution to the field.